# Suppression of IgE-Independent Degranulation of Murine Connective Tissue-Type Mast Cells by Dexamethasone

**DOI:** 10.3390/cells8020112

**Published:** 2019-02-01

**Authors:** Keiko Yamada, Hitomi Sato, Kazuma Sakamaki, Mayumi Kamada, Yasushi Okuno, Nobuyuki Fukuishi, Kazuyuki Furuta, Satoshi Tanaka

**Affiliations:** 1Department of Immunobiology, Division of Pharmaceutical Sciences, Okayama University Graduate School of Medicine, Dentistry, and Pharmaceutical Sciences, Tsushima-naka 1-1-1, Kita-ku, Okayama 700-8530, Japan; satoshi8812@gmail.com (K.Y.); ph422116@s.okayama-u.ac.jp (H.S.); pni477wz@s.okayama-u.ac.jp (K.S.); furutak@okayama-u.ac.jp (K.F.); 2Department of Biomedical Data Intelligence, Graduate School of Medicine, Kyoto University, Shogoin-Kawaharacho, Sakyo-ku, Kyoto 606-8507, Japan; mkamada@kuhp.kyoto-u.ac.jp (M.K.); okuno.yasushi.4c@kyoto-u.ac.jp (Y.O.); 3Department of Pharmacology, College of Pharmacy, Kinjo Gakuin University, 2-1723 Omori, Moriyama-ku, Nagoya, Aichi 463-8521, Japan; nobuf@kinjo-u.ac.jp; 4Department of Pharmacology, Division of Pathological Sciences, Kyoto Pharmaceutical University, Misasagi Nakauchi-cho 5, Yamashina-ku, Kyoto 607-8414, Japan

**Keywords:** mast cell, dexamethasone, trimeric G protein, Mrgpr, skin, inflammation

## Abstract

Steroidal anti-inflammatory drugs are widely used for the treatment of chronic cutaneous inflammation, such as atopic dermatitis, although it remains unknown how they modulate cutaneous mast cell functions. We investigated the effects of prolonged treatment with a synthetic glucocorticoid, dexamethasone, on murine connective tissue-type mast cells using in vitro and in vivo models. Our connective tissue-type bone marrow-derived cultured mast cell model was found to be sensitive to mast cell secretagogues, such as compound 48/80 and substance P, and higher expression levels of α subunit of a trimeric G protein, G_i1_, and several Mas-related G protein-coupled receptor (Mrgpr) subtypes were observed in comparison with immature cultured mast cells. Secretagogue-induced degranulation and up-regulation of these genes was suppressed when cultured in the presence of dexamethasone. The profiles of granule constituents were drastically altered by dexamethasone. Topical application of dexamethasone down-modulated secretagogue-induced degranulation and the expression levels of several Mrgpr subtypes in cutaneous tissue. These results suggest that mast cell-mediated IgE-independent cutaneous inflammation could be suppressed by steroidal anti-inflammatory drugs through the down-regulation of G _αi1_ and several Mrgpr subtypes in mast cells.

## 1. Introduction

Glucocorticoid was found to have the potential to suppress inflammation in 1940s and synthetic glucocorticoids, steroidal anti-inflammatory drugs, have been widely prescribed for the treatment of various chronic inflammatory diseases, such as atopic dermatitis and autoimmune disorders [1,2]. A large part of glucocorticoid-mediated effects arises through its binding to glucocorticoid receptor (GR, Nr3c1). Because GR is ubiquitously expressed in a variety of cells, it is quite difficult to attribute the anti-inflammatory effects of glucocorticoids to the actions on specific cell types. Steroidal anti-inflammatory drugs have been frequently used as a therapeutic intervention for various cutaneous inflammatory diseases, in some of which cutaneous mast cells were found to a play critical role, although it remains to be fully clarified how they act on cutaneous mast cells.

Mast cells originate in the hematopoietic stem cells in the bone marrow and undergo terminal differentiation in the tissue, in which they infiltrate via circulation [3]. These findings indicated that tissue mast cells should have a greater diversity. We previously established a murine bone marrow-derived cultured mast cell model, which had similar characteristics with cutaneous mast cells, through the modification of previous models [4]. We found that this model could undergo degranulation in response to mast cell secretagogues, such as compound 48/80 and substance P. Sensitivity to mast cell secretagogues is one of the signatures of connective tissue-type mast cells. Because no suitable culture models have been developed, the signaling pathways involved in secretagogue-induced degranulation have remained largely unknown [5]. Tatemoto et al. first demonstrated that Mas-related G protein-coupled receptor (Mrgpr) X2 should be involved in secretagogue-induced degranulation of mast cells. Recently, one of the murine orthologues of MrgprX2, MrgprB2, was found to be responsible for pseudo allergic drug responses induced by mast cell degranulation using gene-targeted mice, indicating that the Mrgpr family should be involved in secretagogue-induced mast cell degranulation [6]. MrgprX2 was found to be up-regulated in the cutaneous mast cells of a patient with severe chronic urticaria [7]. Accumulating evidence suggests that IgE-independent activation of mast cells should play a critical role in a wide variety of cutaneous inflammatory diseases; about 20% of patients with atopic dermatitis were reported to have no IgE-sensitization to environmental antigens and low serum IgE levels (intrinsic atopic dermatitis) [8], and about 50% of patients with chronic urticaria were found to spontaneously develop the symptoms [9]. Steroidal anti-inflammatory drugs have been one of the primary therapeutic agents for these inflammatory diseases [10,11,12].

In this paper, we investigated the effects of a synthetic glucocorticoid, dexamethasone, on our connective tissue-type cultured mast cells and on the cutaneous vascular responses in mice in order to clarify how glucocorticoid modulates the function of cutaneous mast cells.

## 2. Materials and Methods

### 2.1. Mice

Specific-pathogen-free, 8 to 10-week-old male BALB/c mice were obtained from Japan SLC (Hamamatsu, Japan), and all mice were kept in a specific-pathogen-free animal facility at Okayama University. This study was approved by the Committee on Animal Experiments of Okayama University (Approved #OKU-2012218, 2015040, and 2015430).

### 2.2. Materials

The following materials were commercially obtained from the sources indicated: Dexamethasone, *p*-nitrophenyl-β-d-2-acetoamide-2-deoxyglucopyranoside, compound 48/80, an anti-dinitrophenyl IgE antibody (clone SPE-7), *N*-succinyl-Ala-Ala-Pro-Phe-*p*NA, mitomycin C, substance P, and dinitrophenyl human serum albumin (DNP-HSA) from Sigma-Aldrich (St. Louis, MO, USA), Toluidine blue, Safranin-O, and Evans blue from Wako Pure Chemical Industries (Osaka, Japan), an anti-trinitrophenyl IgE antibody (clone IgE-3) from BD Biosciences (San Diego, CA, USA), trinitrophenyl bovine serum albumin (TNP-BSA) from LSL (Tokyo, Japan), H-d-Ile-Pro-Arg-*p*NA (S-2288) from Chromogenix (Milano, Italy), *N*-(4-Methoxyphenylazoformyl)-Phe-OH potassium salt (M-2245) from Bachem AG (Bubendorf, Switzerland), an anti-G_αi2_ antibody from Santa Cruz Biotechnology (Dallas, TX, USA), an anti-pan-actin antibody (clone C4), an anti-G_αi1_ antibody, an anti-G_αi3_ antibody, and thapsigargin from Merck Millipore (Billerica, MA, USA), a horseradish peroxidase (HRP)-conjugated anti-mouse IgG antibody and a HRP-conjugated anti-rabbit IgG antibody from Agilent Technologies (Santa Clara, CA, USA) and recombinant mouse IL-3 from R & D Systems (Minneapolis, MN, USA). All other chemicals were commercial products of reagent grade.

### 2.3. Preparation of Bone Marrow-Derived Cultured Mast Cells

Preparation of IL-3-dependent bone marrow-derived cultured mast cells (BMMCs) and connective tissue type mast cell-like cultured mast cells (CTMC-like MCs) was performed as previously described [4]. Briefly, bone marrow cells obtained from male BALB/c mice were cultured in the presence of 10 ng/mL IL-3 for ~30 days. Greater than 95% of the cells exhibited metachromasy by acidic toluidine blue staining and were FcεRI^+^c-kit^+^ on the flow cytometry. CTMC-like MCs were obtained through 16-days of co-culture of BMMCs with mitomycin C-treated Swiss 3T3 fibroblasts in the presence of 100 ng/mL recombinant murine stem cell factor (SCF). Greater than 90% of the cells were confirmed as mature mast cells by Safranin-O staining on day-16. Dexamethasone (final concentration, 1 µM) was added to the culture medium simultaneously with SCF with 48 h of interval thorough the co-culture period.

### 2.4. Measurement of Degranulation

Cultured mast cells were suspended in PIPES-buffer (25 mM PIPES-NaOH, pH 7.4 containing 125 mM NaCl, 2.7 mM KCl, 1 mM CaCl_2_, 5.6 mM glucose, and 0.1% bovine serum albumin), and then stimulated for 30 min at 37 °C. They were centrifuged at 800× *g* at 4 °C for 5 min to obtain the supernatants (extracellular fractions, E). The resultant pellets were resuspended in PIPES-buffer containing 0.5% Triton X-100 and were centrifuged at 10,000× *g* for 10 min to obtain the supernatants (cell-associated fractions, C). Degranulation was evaluated by measuring enzyme activity of a granule enzyme, β-hexosaminidase, in each fraction, using the specific substrate, *p*-nitrophenyl-β-d-2-acetoamide-2-deoxyglucopyranoside (3.4 mM). The reactions were performed in 67 mM citrate, pH 4.5. The amounts of *p*-nitrophenol were determined by measuring OD_405_. The percentages of degranulation were calculated; E/(C + E) × 100 (%).

### 2.5. Measurement of Histamine and IL-6

The amount of histamine was determined by the fluorometrical method with *O*-phthalaldehyde [13]. Tissues were homogenized using a Polytron homogenizer (Kinematica AG, Schweiz, Switzerland) in phosphate buffered saline containing 2 M NaCl and the resultant homogenate was treated with 0.5% Triton X-100. The soluble fraction was subjected to histamine assay. Briefly, samples were deproteinized by adding perchloric acid (fin. 3%) and subjected to high performance liquid chromatography with a cation-exchange column, WCX-1 (Shimadzu, Kyoto, Japan). Post-column derivatizing of histamine by *O*-phthalaldehyde was performed and the resultant products were measured fluorometrically. The amount of IL-6 in the medium was measured using the ELISA system (BD Biosciences, San Diego, CA, USA) according to the manufacturer’s instruction.

### 2.6. Measurement of Granule Protease Activities

Three categories of granule protease activity were measured using their specific substrates as described previously [4]. The cells were washed in phosphate-buffered saline (PBS), incubated in PBS containing 2 M NaCl and 0.5% Triton X-100 at 4 °C for 30 min and then centrifuged at 12,000× *g* at 4 °C for 30 min. The resultant supernatants were subjected to granule protease assays. Chymotryptic activity was measured in 33.3 mM Tris-HCl, pH 8.3 containing 3.3 mM CaCl_2_ and 0.3 mM *N*-succinyl-Ala-Ala-Pro-Phe-*p*NA. Tryptic activity was measured in 33 mM Tris-HCl, pH 8.3 containing 2 mM S-2288. Carboxypeptidase A activity was measured in 33 mM Tris-HCl, pH 7.5 containing 0.6 mM M-2245. The enzymatic activity was determined by measuring OD_405_.

### 2.7. Quantitative PCR Analysis

Messenger RNA levels of various granule proteases and *Mrgpr* gene family were analyzed by quantitative reverse transcription (RT)-PCR with DNase-treated total RNAs. Total RNAs were prepared using NucleoSpin RNA kit (TaKaRa Bio, Kusatsu, Japan). PCR was performed using StepOne Plus (Thermo Fisher Scientific, Waltham, MA, USA) with KOD SYBR qPCR Mix (TOYOBO, Osaka, Japan) or Fast SYBR Green Master Mix (Thermo Fisher Scientific, Waltham, MA, USA) the specific primer pairs (forward, reverse); *Mcpt4*, (5′-CCT TAC ATG GCC CAT CT-3′, 5′-CTT CCC CGG CTT GAT A-3′), *Mcpt5*, (5′-AGA ACT ACC TGT CGG C-3′, 5′-GTC GTG GAC AAC CAA AT-3′), *Mcpt6*, (5′-CTT TGA ACC GGA TCG T-3′, 5′-CTC GTC ATT ATC AAT GTC GC-3′), *Mcpt7*, (5′-AGC TAT GAC ACG AGA AGG-3′, 5′-GCT TAC GGA GCT GTA CT-3′), *Cpa3*, (5′-GAT GTC TCG TGG GAC T-3′, 5′-GCC GTA GAT GTA ACG GG-3′), *Mrgpra4*, (5′-CCT GTG TGC TGT GAT CTG GT-3′, 5′-TCA CGG TTA ATC CAG GGC AC-3′), *Mrgprb1*, (5′-GAC ACA GAG CAA ATT ACC ATC TTC-3′, 5′-CAA GGT TGA GGA TGT AGA CAG AG-3′), *Mrgprb2*, (5’-TGC TTG TCT GTA ATA TGG CCC-3’, 5’-GTC ACA TAC AGC CTG GTC ATA G-3’), *Mrgprb10*, (5′-CCC AGG TTG GTG GAA CTG TT-3′, 5′-GCC AGA AGC CTG ACA GTA GG-3′), *Mrgprc11*, (5′-CTA GCA TCC ACA ACC CCA G-3′, 5′-TGT TTC CTG CCA GTC CAA C-3′), *Mrgpre*, (5′-AGA ACT ACC TGT CGG C-3′, 5′-TTG CCT TCT GGC AGT GAT-3′) and *Gapdh*, (5′-TGT GTC CGT CGT GGA TCT GA-3′, 5′-TTG CTG TTG AAG TCG CAG GAG-3′).

### 2.8. Immunoblot Analyses

Immunoblot analyses were performed as described previously [14]. SDS-PAGE was performed using 10% slab gels and polyvinylidene difluoride membrane transfer was carried out by the semi-dry blotting method. Immunoreactive bands were detected by horseradish peroxidase-conjugated secondary antibodies and were visualized by ECL Western Blotting Detection Reagents (GE healthcare, Chicago, IL). Incubation with the primary antibodies were performed at the following concentrations for 60 min at 37 °C; 1:3000 (G_αi1_), 1:250 (G_αi2_), 1:1000 (G_αi3_), and 1:6000 (actin). Incubation with the secondary antibodies conjugated with HRP were performed at the following concentrations for 60 min at room temperature; 1:3000 (anti-mouse IgG), and 1:3000 (anti-rabbit IgG).

### 2.9. Gene Expression Analysis by Next Generation Sequencing

Sequence reads from each group were individually aligned to the mm10 genome assembly (GRCm38) using TopHat v2.1.0 [15] with default parameters. Aligned read counts were calculated using HTSeq version 0.6.1 [16] with the RefSeq gene annotations obtained from the UCSC Genome Browser [17]. Read counts were then analyzed using DESeq version 1.20.0 [17] to detect differential expression genes (DEGs). Briefly, DESeq normalizes the raw read counts for each sample using size factors, which are calculated from the median of the ratio of observed count to geometric mean for each gene across all samples, and then infers DEGs based on the negative binomial distribution with estimated dispersion and mean linked by local regression. For without biological replicates situation, DESeq estimates dispersion using the samples from the different conditions as replicates. Three comparisons were made between BMMCs and CTMC-like MCs, between BMMC and CTMC-like MCs treated with dexamethasone, and between CTMC-like MCs and CTMC-like MCs treated with dexamethasone. The detected DEGs for each comparison were filtered to those having a false discovery rate (FDR) < 0.1 and divided into up-regulated genes and down-regulated genes based on the logarithmic fold change of normalized counts.

### 2.10. Dexamethasone Treatment

Dexamethasone (20 µL/site, dissolved in acetone) was applied daily to the surface of the ear tissues of mice for 6 days. A series of experiments were performed 24 h after the last application of dexamethasone.

### 2.11. Evaluation of Cutaneous Extravasation Induced upon IgE-dependent Passive Cutaneous Anaphylaxis, and Treatment with Compound 48/80 or Histamine

Mice were intracutaneously sensitized with IgE (30 ng/site, clone SPE-7) in the ear tissue 24 h before the challenge with an intravenous injection of 60 µg DNP-HSA in 0.2 mL saline containing 1 mg Evans blue. The ear tissues were collected 30 min after the challenge and lysed in 3 N KOH. The amount of Evans blue dye was determined by measuring the value of OD_620_. Extravasation responses induced by IgE-independent stimulus were determined by monitoring the dye leakage as described above when non-sensitized male BALB/c mice were intracutaneously injected with compound 48/80 (30 µg/site) or histamine (30 µg/site).

### 2.12. Histological Evaluation of Cutaneous Mast Cells

IgE-mediated antigen challenge and compound 48/80 stimulation were performed as described above without injection of Evans blue dye and the ear tissues were collected 3 min after the stimulation. Cutaneous mast cells were visualized by the acidic toluidine blue staining (pH 3.3) and the numbers of degranulated and intact mast cells were respectively counted.

### 2.13. Statistical Analysis

Data are presented as the means ± SEM. Statistical significance for comparisons was determined using one-way ANOVA. Additional comparisons were made with Dunnett multiple comparison test for comparison with the control groups or Tukey–Kramer multiple comparison test for all pairs of column comparison. Two-tailed unpaired Student’s *t*-test was used for comparison between two populations.

## 3. Results

### 3.1. Characteristic Changes of the Co-Cultured Mast Cells Induced by Prolonged Treatment with Dexamethasone

We first investigated the effects of dexamethasone on proliferation of BMMCs when they were co-cultured with Swiss 3T3 fibroblasts in the presence of SCF. The number of mast cells was increased during the co-culture period, whereas it remained unchanged in the presence of dexamethasone (Figure 1a). A slight decrease in the number of mast cells cultured in the presence of dexamethasone resulted from the loss during the repeated subculture processes. The cell viability was unchanged during the co-culture periods (>95%). Granule maturation was monitored by Safranin staining, which reflects the amount of hyper sulfated proteoglycans, such as heparin, stored in the granules. Dexamethasone did not affect the percentages of Safranin-positive cells and the granule number and morphology (Figure 1b). We previously reported that granule protease activities were drastically increased during the co-culture periods [4]. Prolonged treatment with dexamethasone abolished the induction of chymotryptic activity but enhanced the carboxypeptidase A activity in CTMC-like MCs (Figure 2a,c). The tryptic activity was not changed until day-12 but was significantly decreased at day-16 in the cells co-cultured in the presence of dexamethasone (Figure 2b). Expression of Mcpt4, Mcpt6, and Mcpt7 were all up-regulated in CTMC-like MCs and were suppressed in the presence of dexamethasone, indicating that dexamethasone should affect the granule protease expression at the transcriptional levels, whereas no significant changes were observed in the expression levels of Mcpt5 and Cpa3 (Figure 2d). Cellular histamine content was drastically increased in the presence of dexamethasone (>8 fold at day-16, Figure 3a), which is consistent with a previous study that exhibited the dexamethasone-mediated induction of histidine decarboxylase (HDC), which is the rate-limiting enzyme for histamine synthesis, in a mouse mastocytoma, P-815 [18]. We, indeed, confirmed that the enzymatic activity of HDC increased five-fold in the cells cultured in the presence of dexamethasone (HDC activity at day-4 (nmol/min/mg protein), control, 0.420 ± 0.138, +dexamethasone, 2.11 * ± 0.424, * *p* < 0.05, n = 3). Unexpectedly, enzymatic activity of β-hexosaminidase, a lysosomal enzyme, which might play a critical role in bactericidal action [19] and is often used for monitoring degranulation levels, was significantly up-regulated in CTMC-like MCs obtained in the presence of dexamethasone (Figure 3b).

### 3.2. Suppression of G_i_-Mediated Degranulation in Mast Cells Cultured in the Presence of Dexamethasone

BMMCs co-cultured with Swiss 3T3 fibroblasts were found to undergo degranulation in response to basic secretagogues, such as compound 48/80 and substance P, which is one of the characteristics of CTMCs and is mediated by pertussis toxin-sensitive G_i_ proteins [5]. Degranulation induced by these secretagogues was abolished in the cells co-cultured in the presence of dexamethasone, whereas that upon IgE-mediated antigen stimulation remained unchanged (Figure 3c,d). The suppressive effects of dexamethasone on the G_i_-dependent degranulation were not observed when dexamethasone was added 24 h before stimulation (Figure 3f), indicating that dexamethasone-mediated suppression of G_i_-dependent degranulation requires long-term characteristic changes.

Previous studies demonstrated that treatment with dexamethasone down-modulated the surface expression of FcεRI and thereby suppressed antigen-induced degranulation in mast cells [20,21]. However, no significant changes were observed in the levels of degranulation of activated BMMCs, which were treated with dexamethasone 24 h before the stimulation (Figure 3g,h). Surface expression levels of FcεRI were comparable between the control and dexamethasone-treated BMMCs (mean fluorescent intensity; control, 47.2 ± 4.01, +dexamethasone, 41.8 ± 0.768, n = 3), whereas those of c-kit were significantly decreased in the dexamethasone-treated cells (mean fluorescent intensity; control, 104 ± 3.55, +dexamethasone, 81.5 * ± 0.379, n = 3, * *p* < 0.05). In contrast to unchanged levels of degranulation, treatment with dexamethasone for 24 h significantly suppressed antigen-induced IL-6 production whereas Ca^2+^ influx-induced IL-6 production was unchanged in the presence of dexamethasone (Figure 3i,j).

Because G_αi1_ protein was found to be up-regulated during the co-culture period [4], we then investigated the effects of dexamethasone on the expression of three subtypes of α subunit of trimeric G_i_. Expression of G_αi1_ protein was exclusively induced during the co-culture period as previously reported and was significantly suppressed by prolonged treatment with dexamethasone, whereas no obvious changes in the other G_αi_ protein expression were observed (Figure 4a–d). Recently, the possible candidates for the receptors of mast cell secretagogues have been identified; various secretagogues were found to act as the agonists of MRGPRX2 [6], and one of its murine orthologues, MrgprB2, was identified as the primary receptor for various mast cell secretagogues using the gene targeted mice [7]. We investigated the expression levels of six murine Mrgpr family genes based on the results obtained through next generation sequencing analysis. Messenger RNA expression of Mrgpra4, b1, b2, b10, and e were detected in the cultured mast cells and murine peritoneal cells. Mrgprc11 was not detected in murine peritoneal cells but expressed in the cultured mast cells. Mrgprb1, b2, b10, and c11 were found to be up-regulated during the co-culture period, whereas Mrgpra4 and e were down-regulated (Figure 5). Overall, the presence of dexamethasone during the co-cultured period suppressed mRNA expression of these Mrgpr genes.

### 3.3. Effects of Dexamethasone on Gene Expression Profiles of Cultured Mast Cells

We then investigated the gene expression profiles by next generation sequencing analysis. We extracted 376 genes as the up-regulated genes (FDR <0.1) in CTMC-like MCs in comparison with BMMCs. This population included the characteristic genes of CTMCs, which we previously identified by microarray analyses [4], such as Bgn, Cd81, Gnai1, Icam1, Mcpt4, Ptges, Ptgis, Ptgs2, Thbs1, and Tpsab1. Among them, 113 genes were not induced in the cells cultured in the presence of dexamethasone (Figure 6, Group I and III). These dexamethasone-sensitive groups included genes, such as Cd81, Gnai1, Icam1, Mcpt4, Mmp9, Nfkbia, and Tpsab1. The number of genes, which were up-regulated in CTMC-like MCs but were insensitive to dexamethasone, was 263 (Figure 6, Group II). They included genes involved in arachidonic acid metabolism, such as Ptges, Ptgs2, and Ptgis.

### 3.4. Effects of Dexamethasone on Murine Cutaneous Vascular Responses

We then investigated the effects of dexamethasone on cutaneous vascular responses. Six days of topical application of dexamethasone did not affect the number of cutaneous tissue mast cells, whereas cutaneous histamine content was significantly increased (Figure 7a,b). Four *Mrgpr* genes, Mrgpra4, b2, b10, and e, were expressed in the ear tissue, whereas the expression of Mrgprb1 and c11 could not be detected by RT-PCR (Figure 7c). Prolonged treatment with dexamethasone down-regulated all Mrgpr genes expressed in the ear tissue.

Extravasation responses evaluated by Evans blue dye leakage upon IgE-mediated antigen stimulation or compound 48/80 were significantly attenuated in the mice treated daily with 0.05% dexamethasone (Figure 8a,b). We then investigated the sensitivity to histamine, which is the major vasoactive mediator derived from mast cells, and found that histamine-induced extravasation response was also attenuated in the dexamethasone-treated mice (Figure 8c). These findings implied that decreased dye leakages observed in the dexamethasone-treated mice might result from impaired vascular responses to histamine rather than impaired degranulation. We, therefore, assessed the frequencies of degranulation of tissue mast cells by histological analyses with acidic toluidine blue staining. In the mice treated daily with 0.05% dexamethasone, the number of degranulated mast cells was slightly but significantly increased upon IgE-mediated antigen stimulation, whereas that was significantly decreased upon the compound 48/80 application (Figure 8d,e). Suppression of compound 48/80-induced degranulation was more pronounced in the mice treated daily with 0.2% dexamethasone, whereas antigen-induced degranulation was also suppressed under this condition (Figure 8f,g).

## 4. Discussion

We demonstrated here that prolonged treatment with a synthetic glucocorticoid, dexamethasone, could suppress secretagogue-induced degranulation of murine mast cells. Because glucocorticoid receptor is ubiquitously expressed, it remains unknown how glucocorticoid suppress cutaneous inflammation. Our findings strongly suggest that the down-regulation of *Gnai1* and *Mrgpr* family in mast cells should be involved in impaired secretagogue-induced degranulation. A large part of secretagogue-induced degranulation was found to be sensitive to pertussis toxin [5]. Aridor et al. suggested that G_i3_ should be involved in secretagogue-induced degranulation of rat peritoneal mast cells [22]. It is likely that cooperation between several Mrgpr subtypes and G_i1_ should be responsible for secretagogue-induced degranulation of murine mast cells. This mechanism might account, at least in part, for the therapeutic effects of steroidal anti-inflammatory drugs on IgE-independent cutaneous inflammation including intrinsic atopic dermatitis and contact dermatitis.

We found here that prolonged treatment with dexamethasone impaired the vascular responses in the cutaneous tissues. Because glucocorticoids were found to augment the functions of tight junctions of vascular endothelial cells [23,24], lower sensitivity to histamine of the cutaneous endothelial cells in mice treated with dexamethasone might result from enhanced tight junction integrity. Cutaneous vascular responses induced by compound 48/80 might be strongly suppressed by dexamethasone through its effects both on endothelial cells and on mast cells.

The effects of glucocorticoids on degranulation induced by IgE-mediated antigen stimulation have been extensively investigated. A majority of studies demonstrated that dexamethasone could significantly suppress degranulation of mast cells upon antigen stimulation [20,21,25,26,27]. It remains to be determined how dexamethasone should act on the signaling pathways involved in degranulation. Some of these studies suggested that the surface expression of FcεRI was decreased in the presence of dexamethasone. We observed here that prolonged treatment with dexamethasone did not affect antigen-induced degranulation of cultured mast cells but significantly suppressed that of cutaneous mast cells. We could not reproduce the previous findings that 24-h of treatment with dexamethasone suppressed antigen-induced degranulation of BMMCs [20,21]. Antigen-induced IL-6 release was abolished by 24-h of treatment with dexamethasone, excluding the possibility that BMMCs should be insensitive to glucocorticoids in our system. We have no good explanation for this discrepancy, although the methods for preparation of BMMCs are slightly different from our model.

It remains controversial how glucocorticoid affects the number of mast cells. Eklund et al. reported that dexamethasone suppressed proliferation of murine BMMCs induced by SCF or IL-3, which is consistent with our findings [28]. Methylprednisolone could deplete intestinal mast cells but did not affect the number of connective tissue-type mast cells in rats [29]. In cutaneous tissues, the effects of glucocorticoids on fibroblasts might be involved in the regulation of the mast cell number. Chronic topical treatments with fluocinonide decreased the number of cutaneous mast cells through the down-regulation of SCF in the fibroblasts, which induced apoptosis of the mast cells [30]. It is likely that the expression levels of SCF in Swiss 3T3 were down-regulated in the presence of dexamethasone in our system. However, the presence of a large amount of soluble exogenous SCF may prevent apoptotic cell death of CTMC-like MCs. Clinical studies demonstrated that prolonged treatment with clobetasol-17-propionate and fluocinonide could down-modulate the number of cutaneous mast cells [31,32]. Because no changes in the number of cutaneous mast cells were observed in our system, dexamethasone might not affect the viability of them in the range of concentrations used here. We used Swiss 3T3 fibroblasts as the feeder cells to prepare CTMC-like MCs, raising the possibility that dexamethasone should affect the phenotype of mast cells indirectly through the effects on Swiss 3T3 cells. It is quite difficult to distinguish the direct effects from those through the fibroblasts. Kusunose et al. reported that treatment of murine fibroblasts with mitomycin c should attenuate the nuclear translocation of GR [33]. Because Swiss 3T3 cells were pretreated with mitomycin c before the co-culture in our system, the effects of dexamethasone on them may be relatively small.

We observed here that granule enzymes, such as carboxypeptidase A and β-hexosaminidase, and histamine contents were significantly up-regulated in the cultured mast cells generated in the presence of dexamethasone. Increase in histamine content was also observed in the cutaneous tissues of mice treated with dexamethasone. It is plausible that dexamethasone should act on the cutaneous mast cells and augmented the cellular histamine content, because a majority of cutaneous histamine was found to be derived from mast cells in mice and it is unlikely that prolonged treatment with dexamethasone induced infiltration of the other kinds of histamine-forming cells, such as activated macrophages and neutrophils [34]. Accumulating evidence suggests that glucocorticoids could not only suppress inflammatory responses but also enhance innate immune responses [35,36]. Carboxypeptidase A and a granule proteoglycan core, serglycin, were found to be up-regulated in BMMCs treated with dexamethasone [28]. Mast cells were found to contribute to wound healing through release of their mediators including histamine, which were involved in proliferation of fibroblasts and angiogenesis [37,38,39]. Because accumulating evidence suggests that mast cell proteases should often play pro-inflammatory roles [40], dexamethasone-mediated down-modulation of these proteases may attenuate the pro-inflammatory impact of mast cells. Phenotypic changes of cutaneous mast cells induced by prolonged treatment with dexamethasone may contribute to its beneficial effects on wound healing.

We found that a variety of murine Mrgpr family was expressed in cultured mast cells in addition to *Mrgprb2* and that their expression levels were dynamically changed. Although McNeil et al. highlighted the critical role of MrgprB2 using gene targeted mice [7], it is plausible that the other Mrgpr family should be involved in IgE-independent degranulation of mature mast cells in cooperation with MrgprB2. Mrgpr family may respond to various secretagogues through its heterodimerization. We also detected mRNA expression of Mrgpr family in murine cutaneous tissues, the levels of which were decreased in the presence of dexamethasone. Because previous studies indicated that Mrgpr family was expressed exclusively in the sensory nerve, of which cell body is localized in the dorsal root ganglion, except that mast cells expressed a part of it, mRNA expression of Mrgpr subtypes in the cutaneous tissues might indicate that cutaneous mast cells should express them. Characterization of these Mrgpr subtypes in addition to MrgprB2 is necessary for understanding the mechanism of IgE-independent inflammatory responses.

## Figures and Tables

**Figure 1 cells-08-00112-f001:**
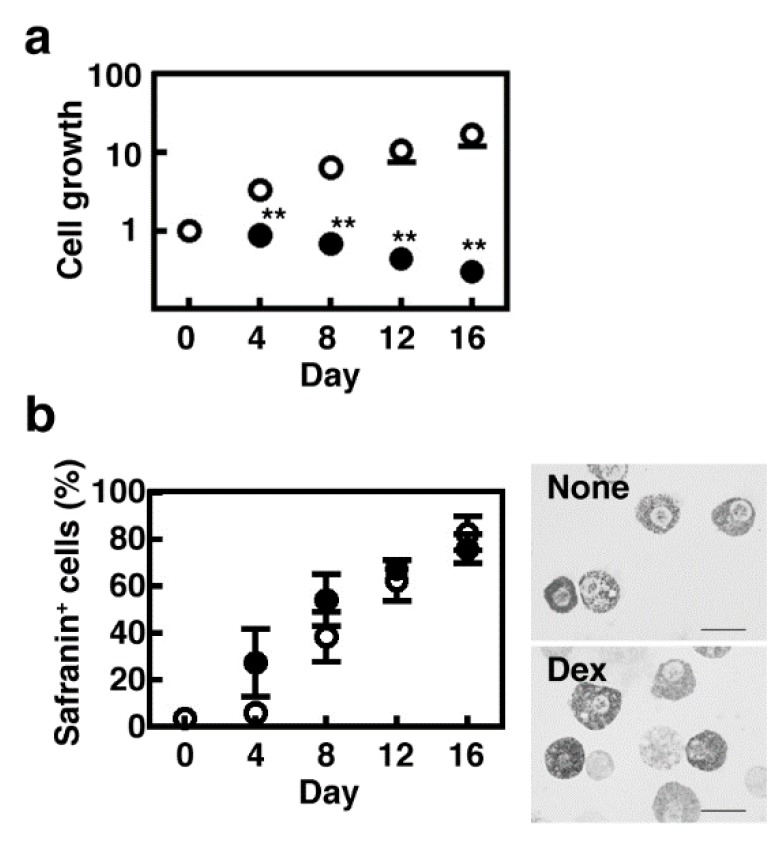
Bone marrow-derived cultured mast cells (BMMCs) were co-cultured with Swiss 3T3 fibroblasts in the presence (closed circles) or absence (open circles) of 1 µM dexamethasone for 16 days as described in Materials and Methods. (**a**) The numbers of the cultured mast cells were counted on day-0, 4, 8, 12, and 16. Values are presented as the means ± SEMs (n = 4). The values ** *p* < 0.01 are regarded as significant. (**b**) The ratios of the Safranin-positive cells were determined. Values are presented as the means ± SEMs (n = 4).

**Figure 2 cells-08-00112-f002:**
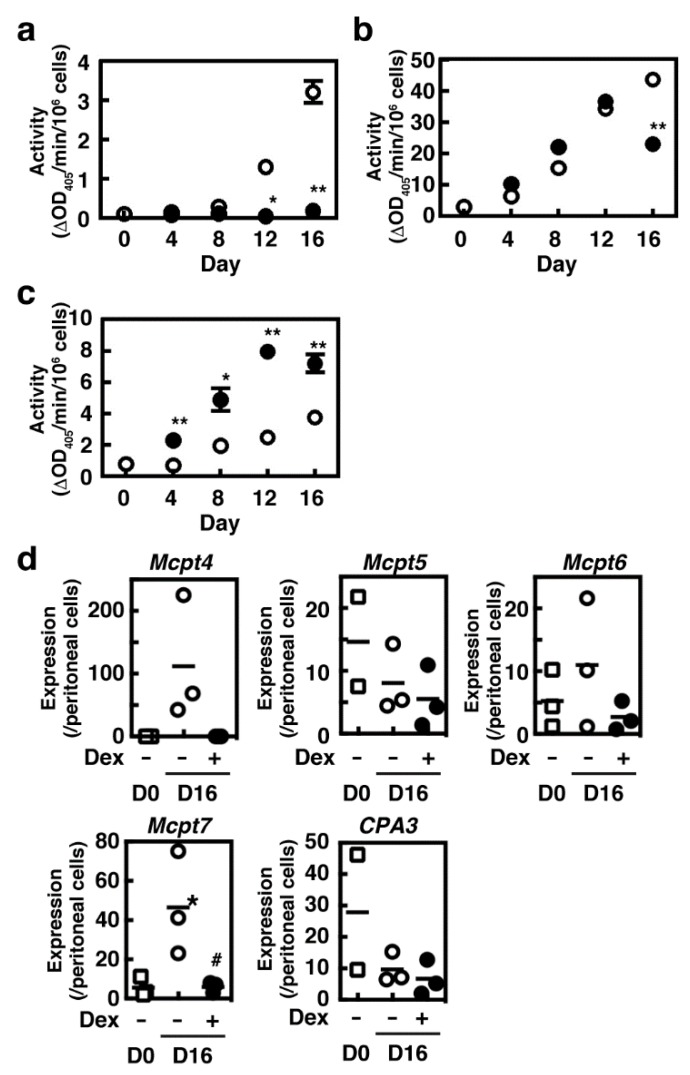
BMMCs were co-cultured with Swiss 3T3 fibroblasts in the presence (closed circles or columns) or absence (open circles or columns) of 1 µM dexamethasone for 16 days as described in Materials and Methods. (**a**–**c**) Enzymatic activities of three kinds of granule proteases (**a**); chymotryptic activity, (**b**); tryptic activity, and (**c**); carboxypeptidase A activity) were measured. Values are presented as the means ± SEMs (n = 3). Values with * *p* < 0.05 and ** *p* < 0.01 are regarded as significant. (**d**) Expression levels of granule protease genes (*Mcpt4*, *5*, *6*, *7*, and *CPA3*) were determined by quantitative RT-PCR analyses. Values with * *p* < 0.05 (vs. D0) and ^#^
*p* < 0.05 (vs. D16, (−)Dex) are regarded as significant.

**Figure 3 cells-08-00112-f003:**
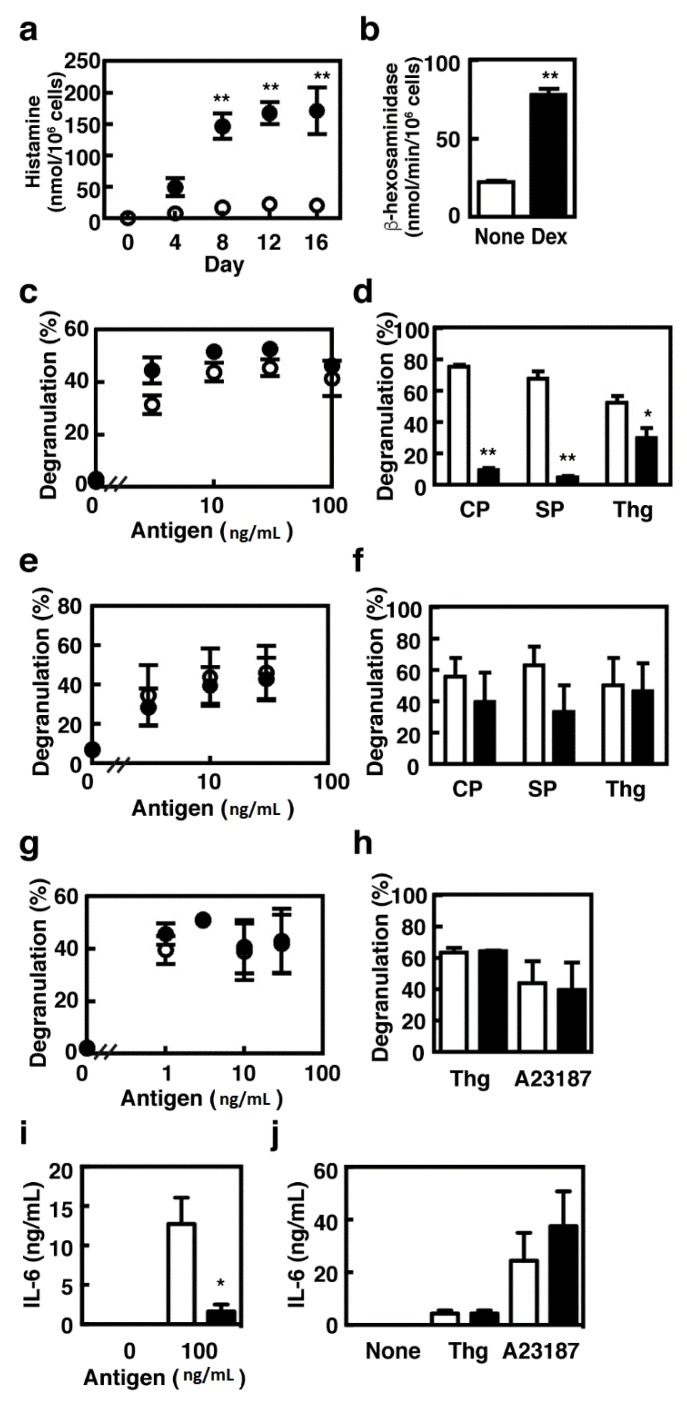
(**a**,**b**) The cellular histamine contents and enzymatic activities of β-hexosaminidase in the mast cells co-cultured for 16 days in the presence (closed circles) or absence (open circles) of 1 µM dexamethasone were measured. (**c**–**f**) The co-cultured mast cells were sensitized with IgE (1 µg/mL, clone IgE-3) for 3 h and then stimulated with the indicated concentrations of the antigen, or stimulated with compound 48/80 (CP, 10 µg/mL), substance P (SP, 100 µM), or thapsigargin (Thg, 300 nM) without sensitization. Degranulation upon IgE-mediated antigen stimulation (**c**) and treatment with compound 48/80, substance P, or thapsigargin (**d**) was measured in the mast cells co-cultured for 16 days in the presence (closed circles or columns) or absence (open circles or columns) of 1 µM dexamethasone. (**e**,**f**) BMMCs were co-cultured for 16 days and were treated with 1 µM dexamethasone during the last 24 h (closed circles and columns). Degranulation was then measured as described above. (**g**–**j**) BMMCs were treated without (open circles or columns) or with 1 µM dexamethasone (closed circles or columns) for 24 h. The cells were then sensitized with 1 µg/mL IgE (clone IgE-3) for 3 h and stimulated with the indicated concentrations of the antigen or stimulated with thapsigargin (Thg, 300 nM) or A23187 (A23187, 1 µM). Degranulation (**g**,**h**) and IL-6 release (**i**,**j**) were measured. The degree of degranulation was determined by measuring β-hexosaminidase activity. Values are presented as the means ± SEMs (n = 3). Values with * *p* < 0.05 and ** *p* < 0.01 are regarded as significant.

**Figure 4 cells-08-00112-f004:**
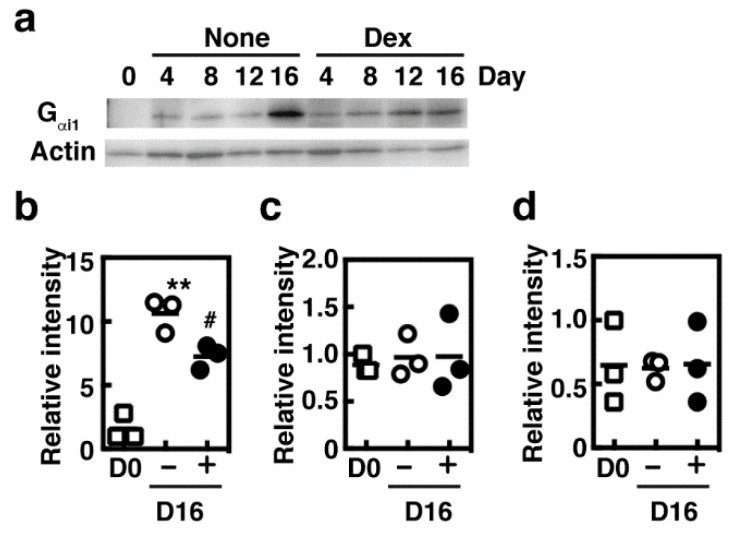
BMMCs were co-cultured with Swiss 3T3 fibroblasts in the presence (Dex) or absence (None) of 1 µM dexamethasone for 16 days as described in Materials and Methods. (**a**) Expression of G_αi1_ was visualized by immunoblot analyses using an anti-G_αi1_ antibody. Expression of actin was measured as the loading control. (**b**–**d**) Expression levels of various G_αi_ proteins (**b**; G_αi1_, **c**; G_αi2_, and **d**; G_αi3_) in BMMCs (D0, open squares), and the day-16 co-cultured mast cells prepared in the presence (+, closed circles) or absence (−, open circles) of 1 µM dexamethasone were densitometrically determined (n = 3). Values with ** *p* < 0.01 (vs. D0) and ^#^
*p* < 0.05 (vs. D16, (−)Dex) are regarded as significant.

**Figure 5 cells-08-00112-f005:**
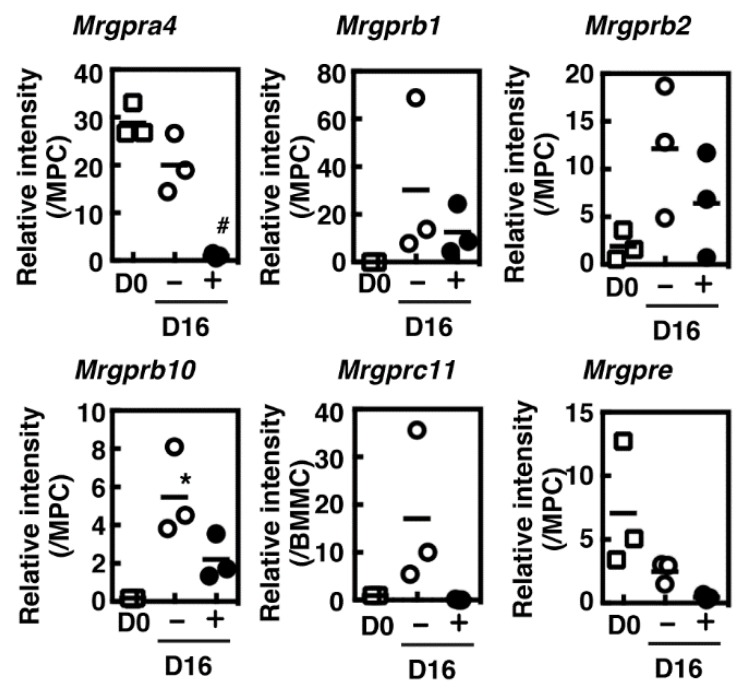
BMMCs were co-cultured with Swiss 3T3 fibroblasts in the presence (Dex) or absence (None) of 1 µM dexamethasone for 16 days as described in Materials and Methods. Expression levels of *Mrgpr* family genes in BMMCs (D0, open squares), and the day-16 co-cultured mast cells prepared in the presence (+, closed circles) or absence (−, open circles) of 1 µM dexamethasone were measured using quantitative RT-PCR. Relative expression levels were calculated based on the expression levels in mouse peritoneal cells (MPC) or those in BMMCs (BMMC) (n = 3). Values with * *p* < 0.05 (vs. D0) and ^#^
*p* < 0.05 (vs. D16, (−)Dex) are regarded as significant.

**Figure 6 cells-08-00112-f006:**
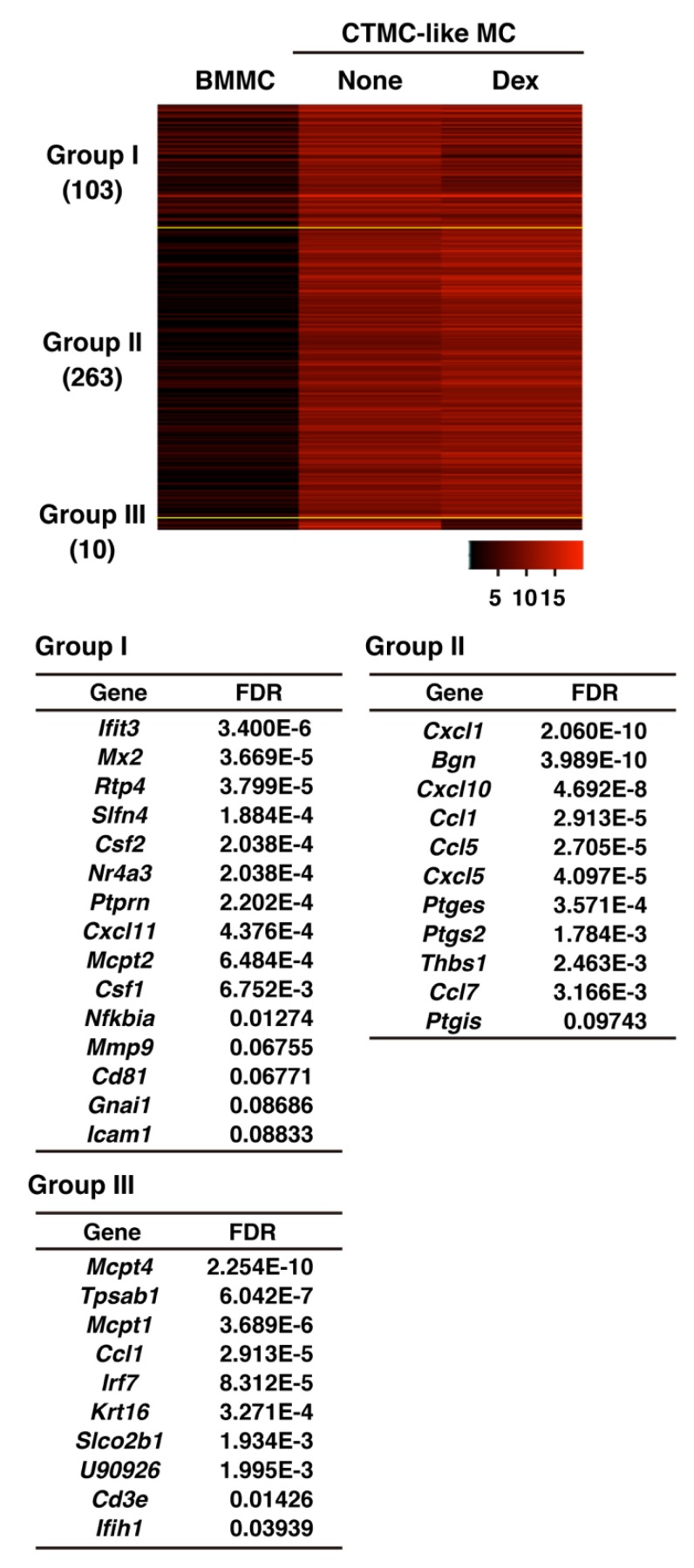
Each RNA sample was collected from the cultured mast cells (BMMCs, connective tissue type mast cell-like cultured mast cells (CTMC-like MCs), and CTMC-like MCs prepared in the presence of Dex) and gene expression analyses were performed by next generation sequencing. Differentially expressed genes between BMMCs and CTMC-like MCs were extracted with the false discovery rate (FDR) <0.1. The heat map presents the expression profiles of the genes, of which expression was increased in CTMC-like MCs. The extracted genes are classified into three clusters based on the expression patterns.

**Figure 7 cells-08-00112-f007:**
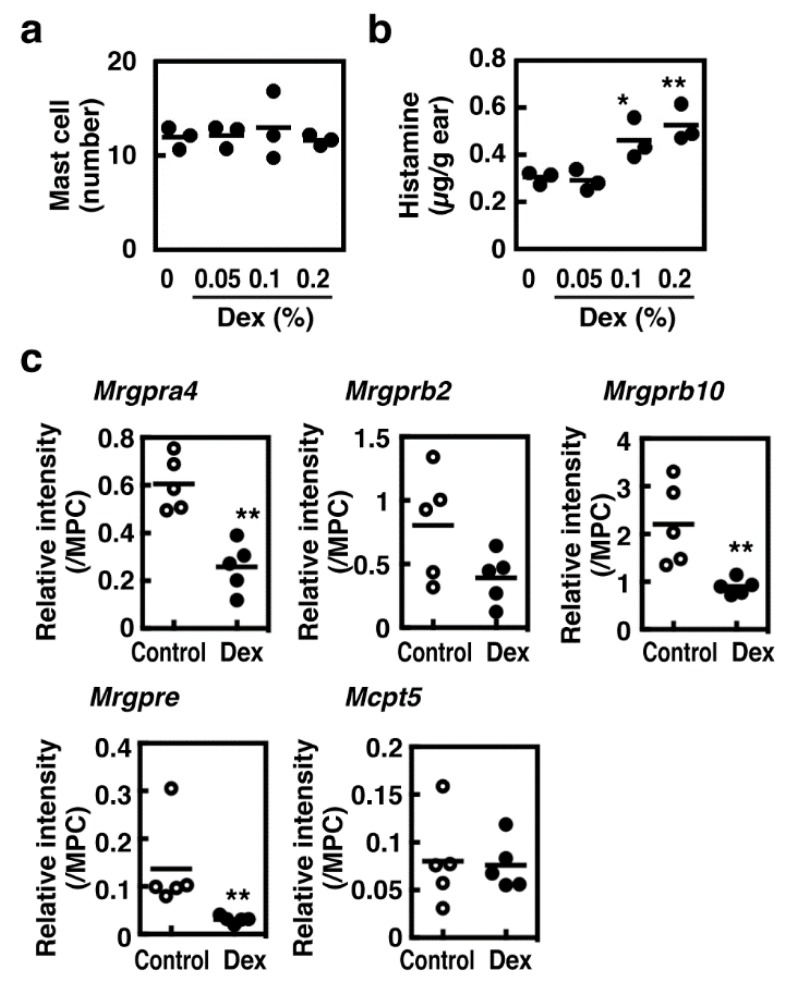
(**a**,**b**) The indicated concentrations of dexamethasone (20 µL/site, dissolved in acetone) were applied daily to the surface of the ear tissue for 6 days. The ear tissues were collected 24 h after the last administration and the numbers of cutaneous mast cells (**a**) and tissue histamine content (**b**) were measured (n = 3). Values with * *p* < 0.05 and ** *p* < 0.01 are regarded as significant. (**c**) Expression levels of *Mrgpr* family genes and *Mcpt5* in the ear tissue were measured using quantitative RT-PCR (Control, open circles; Dex, closed circles). Relative expression levels were calculated based on the expression levels in mouse peritoneal cells (MPC) (n = 5). Values with ** *p* < 0.01 and * *p* < 0.05 are regarded as significant.

**Figure 8 cells-08-00112-f008:**
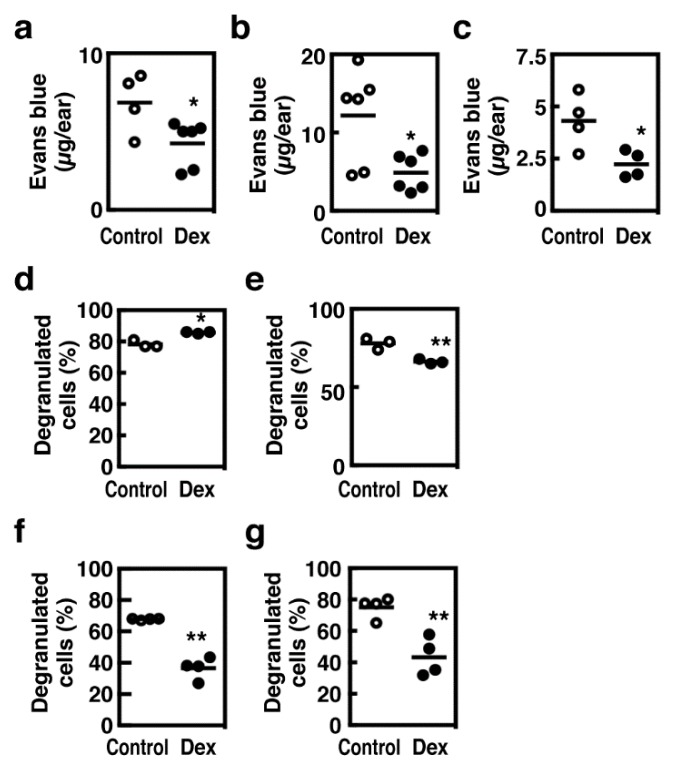
(**a**–**c**) After the daily application of 0.05% dexamethasone to the ear tissue for 6 days, extravasation in the ear tissue was evaluated by Evans blue dye leakage 24 h after the last application (Control, open circles; Dex, closed circles). Mice were subjected to IgE-dependent passive cutaneous anaphylaxis (PCA) reactions (**a**), and intracutaneous injections of compound 48/80 (**b**, 30 µg/site) or histamine (**c**, 30 µg/site). Values with * *p* < 0.05 are regarded as significant. (**d**–**g**) Mice were treated daily without (control) or with 0.05% (**d**,**e**) or 0.2% (**f**,**g**) of dexamethasone (Dex, 20 µl/site, dissolved in acetone) on the surface of ear tissue for 6 days. Mice were then subjected to IgE-dependent PCA reactions (**d**,**f**) or an intracutaneous injection of compound 48/80 (**e**,**g**, 30 µg/site). The ear tissues were collected 3 min after the stimulation and the degrees of degranulation of cutaneous mast cells were determined based on the acidic Toluidine blue staining. Values with * *p* < 0.05 and ** *p* < 0.01 are regarded as significant.

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
