# Peer review of "Suppression of IgE-Independent Degranulation of Murine Connective Tissue-Type Mast Cells by Dexamethasone"

_cells, 2019, doi:10.3390/cells8020112_

Reviewer 1 Report

In the paper titled “Suppression of IgE-independent degranulation of murine connective tissue-type mast cells by  dexamethasone” the Authors investigate, both in vitro and in vivo, the effects of a synthetic glucocorticoid, dexamethasone, on connective tissue-type cultured mast cells obtained by co-culture of  bone marrow-derived mast cells with Swiss 3T3 fibroblasts in the presence of stem cell factor and on the cutaneous vascular responses in mice in order to clarify  how glucocorticoid modulate the functions of cutaneous mast cells.

Since connective tissue-type cultured mast cells  are sensitive to secretagogues, such as compound 48/80 and substance P, they used 48/80 as degranulation inducer

This process was accompanied by up-regulation of a subunit of a trimeric G protein, Gi1, and several  Mas-related G protein-coupled receptor (Mrgpr) subtypes. They also observed that the profiles of granule constituents were drastically  altered by dexamethasone treatment and  several Mrgpr subtypes resulted diminished. In addition, they observed that the number of degranulated cutaneous mast cells in response to compound 48/80 was decreased in mice treated with dexamethasone.

Overall, these results suggest that mast cell-mediated  IgE-independent cutaneous inflammation could be suppressed by steroidal anti-inflammatory  drugs through down-regulation of Gai1 and several Mrgpr subtypes in mast cells.

The work presents many critical points that do not allow it to be accepted in this form. Following are some comments:

The English requires a careful review by a native speaker.

Abstract.

The text is a bit confused in its parts. The authors should more clearly define the objectives, the two experimental models used with relative treatments and results, so that the reading of the summary is more understandable.

Par. 2.3 line 103: Please justify the choice  of the indicated concentration of dexamethasone (1mm) Par. 2.4 Measurement of degranulation:  The authors must detail the experimental procedure, in fact it is not clear through which method the degranulation is quantitatively evaluated.

Par. 2.5 Measurement of histamine and IL-6: Please detail the experimental procedures for the quantitative determination of histamine and IL-6.

Par. 2.6: the same as above.

Par. 2.8. The procedure reported is too synthetic, therefore it is necessary to detail the executive parts of the western, including the primary and secondary antibodies utilized and their relative dilution, other that time of incubation.

2.10. specify the reasons for using 20 ml of dexamethasone.

2.11. the authors have to report the times and the concentrations used for these experimental evaluations, as well as the number of animals used for each test. Report this last data also in the corresponding figures.

 Results:

Figs 2, 4, 5, 7: Please indicate the number of the experiments performed for which the statistical analysis was done.

Discussion

1) Pag 14 line387: Authors should comment on what biological actions would result in IL-6 releasing. In the manuscript text there is no mention in any of its parts

2) The author reported in the abstract that “The numbers of degranulated cutaneous mast cells in response to compound 48/80  were decreased in mice treated with dexamethasone.”

Whereas in discussion section, line 402 they reported that “Because no changes in the number of cutaneous mast cells were observed in our system..”: there seems to be a contradiction.

3) Another point of discussion that needs to be further explored by the authors is this: dexamethasone seems not to act on the number of cells, but rather on the amount of histamine released. Which more comprehensive justification of this phenomenon?

4) Furthermore, the authors could argue how the increase in histamine could be associated with a therapeutic action. Could they provide more exhaustive explanations in this regard to justify the repairing properties of histamine?

 Finally, how do you explain the increased release of histamine unlike other degranulation products? Is it possible that there are multiple mechanisms with which the dexamethason acts or the effect could be linked to a cell heterogeneity?

Author Response

We very much appreciate your critical and insightful comments, which should enhance the accuracy and significance of this manuscript.

Abstract.
Q: The text is a bit confused in its parts. The authors should more clearly define the objectives, the two experimental models used with relative treatments and results, so that the reading of the summary is more understandable.

A: We revised the text of Abstract according to your suggestion. We hope that we could adequately respond to your comments. (Abstract lines 22-31)

Q: Par. 2.3 line 103: Please justify the choice of the indicated concentration of dexamethasone (1 μM)

A: Concentrations of dexamethasone used in previous studies have usually ranged from 1 nM to 10 μM. Although a majority of studies demonstrated that 1 μM of dexamethasone could suppress degranulation induced by IgE-mediated antigen stimulation, Wersil et al. reported that 1 μM of dexamethasone did not suppress antigen-induced degranulation in murine mast cell line, C57 (References #21, #25, and #26). We investigated the effects of dexamethasone at 1 μM in this study in order to obtain reproducible results. Preliminary investigations indicated that some indices showed similar tendencies in the cells treated with 10 nM.

Q: Par. 2.4 Measurement of degranulation:  The authors must detail the experimental procedure, in fact it is not clear through which method the degranulation is quantitatively evaluated.

A: We evaluated the levels of degranulation by measuring the enzymatic activity of a lysosomal enzyme, β-hexosaminidase. We added the detailed procedures of β-hexosaminidase assay. (2.4, lines 123-132)

Q: Par. 2.5 Measurement of histamine and IL-6: Please detail the experimental procedures for the quantitative determination of histamine and IL-6.

A: We added the details of measurement of histamine. We used an ELISA kit to measure the amount of IL-6 in the medium. The experimental procedures of this kit are available on the web site and are similar to the standard ELISA method. (2.5, lines 138-142)

Q: Par. 2.6: the same as above.

A: We added the details of measurement of enzymatic activities of granule proteases. (2.6, lines 147-153)

Q: Par. 2.8. The procedure reported is too synthetic, therefore it is necessary to detail the executive parts of the western, including the primary and secondary antibodies utilized and their relative dilution, other that time of incubation.

A: We added the detailed information of immunoblot analyses. (2.8, lines 181-184)

Q: 2.10. specify the reasons for using 20 μl of dexamethasone.

A: This volume (20 μl) has frequently used in the experiments of contact dermatitis because one could paint the solvent all over the surface of pinna of ear.

Q: 2.11. the authors have to report the times and the concentrations used for these experimental evaluations, as well as the number of animals used for each test. Report this last data also in the corresponding figures.

A: This section describes the experimental conditions in Figure 8a-8c. We described the concentrations of dexamethasone and the number of animals in the legend to Figure 8.

Results:
Q: Figs 2, 4, 5, 7: Please indicate the number of the experiments performed for which the statistical analysis was done.

A: We added the number of experiments in the legends to these Figures. Statistical analyses were performed according to the section 2.13. (Fig. 2, 4, 5, and 7)

Discussion
1) Pag 14 line387: Authors should comment on what biological actions would result in IL-6 releasing. In the manuscript text there is no mention in any of its parts

We investigated the effects of dexamethasone on antigen-induced IL-6 production in order to exclude the possibility that BMMCs were totally insensitive to glucocorticoids. Murine mast cells were found to release IL-6 in response to antigen stimulation but it remains to be clarified in detail how cross-linking of FcεRI should lead to transcriptional activation of IL-6 genes in mast cells, whereas antigen-induced IL-6 release from mast cells was reported to be sensitive to glucocorticoid (Leal-Bermen et al., J. Immunol., 154, 4759-4767, 1995, Kimata et al., Pharmacology, 62, 17-22, 2001).

2) The author reported in the abstract that “The numbers of degranulated cutaneous mast cells in response to compound 48/80 were decreased in mice treated with dexamethasone.”
Whereas in discussion section, line 402 they reported that “Because no changes in the number of cutaneous mast cells were observed in our system..”: there seems to be a contradiction.

We revised the abstract. We intended to describe that the number of cutaneous mast cells that underwent degranulation in response to compound 48/80 was decreased in mice receiving dexamethasone whereas the total cutaneous mast cell number was unchanged.

3) Another point of discussion that needs to be further explored by the authors is this: dexamethasone seems not to act on the number of cells, but rather on the amount of histamine released. Which more comprehensive justification of this phenomenon?

We observed in the co-culture model that the amounts of histamine per cell were significantly increased in the cultured mast cells in the presence of dexamethasone. We speculate that similar increases in the amount of histamine should occur in the cutaneous mast cells in mice treated with dexamethasone, which might lead to the increased histamine content in the cutaneous tissues. Accumulating evidence suggests that histamine in murine cutaneous tissues should come from mast cells (Yamatodani et al., Biochem. Pharmacol. 31, 305-309, 1982, and Ref #34; Taguchi et al., Proc. Natl. Acad. Sci. USA, 79, 6837-6841, 1982). We added the descriptions in the section of discussion. (Discussion, lines 459-463)

4) Furthermore, the authors could argue how the increase in histamine could be associated with a therapeutic action. Could they provide more exhaustive explanations in this regard to justify the repairing properties of histamine?
Finally, how do you explain the increased release of histamine unlike other degranulation products? Is it possible that there are multiple mechanisms with which the dexamethasone acts or the effect could be linked to a cell heterogeneity?

Your critical comments have made us notice the physiological significance of increased histamine. We appreciate your suggestion. Previous studies demonstrated that histamine should be involved in proliferation of fibroblasts and angiogenesis, both of which are essential for wound healing. Because mast cell proteases were found to play pro-inflammatory roles in various diseases, dexamethasone-mediated down-modulation of these proteases may attenuate the pro-inflammatory impact of mast cells. Together with the finding that vascular responses in response to histamine in mice treated with dexamethasone, dexamethasone may change the cutaneous environment in favor of wound healing. We added the explanations in the section of discussion. (Discussion, lines 467-472)

Reviewer 2 Report

Yamada et al show that treatment with dexamethasone suppresses compound 48/8-induced degranulation of mouse connective tissue-type mast cells presumably through down-regulation of Gαi1 and several Mrgpr subtypes in mast cell differentiation. The experiments are convincing and well performed, and the results provide valuable information to the researchers in the relevant fields. However, there are several concerns which should be addressed. 

1.   As the authors mentioned, the in vivo experimental results (Fig. 8, a-c) suggested that decreased leakage of dyes in dexamethasone-treated mice might be due to the decreased vascular permeability. This point should be more deeply discussed, to address the effect of in vivo dexamethasone treatment on compound 48/80-mediated inflammation.

2.   Comparative gene expression profiles between CTMC-like MC and peritoneal MC and/or between peritoneal MC treated with or without dexamethasone will make clear the effects of dexamethasone treatment on the differentiation of connective tissue mast cells.

 3.  To examine the effects of specific knockdown of Mrgpra4, Mrgprb1, Mrgprb2, Mrgprb10,  Mrgpre, Gai1, Gai2, or Gai3 on compound 48/80-mediated degranulation of CTMC-like MC generated by the authors will lead you to make a definite conclusion.

Author Response

We very much appreciate your positive evaluation of our study and constructive comments, which encourage us to further explore how glucocorticoids modulate tissue mast cell functions.

1.   As the authors mentioned, the in vivo experimental results (Fig. 8, a-c) suggested that decreased leakage of dyes in dexamethasone-treated mice might be due to the decreased vascular permeability. This point should be more deeply discussed, to address the effect of in vivo dexamethasone treatment on compound 48/80-mediated inflammation.

According to your suggestion, we further discussed the effects of dexamethasone on vascular responses. In the cutaneous tissues, dexamethasone might have at least two targets, one is vascular endothelial cells and the other is mast cells. Increased amounts of histamine were stored in the cutaneous mast cells in mice treated with dexamethasone, but its release and actions on the neighboring endothelial cells were restricted. We added the explanations in the section of discussion. (Discussion, lines 417-422)

2.   Comparative gene expression profiles between CTMC-like MC and peritoneal MC and/or between peritoneal MC treated with or without dexamethasone will make clear the effects of dexamethasone treatment on the differentiation of connective tissue mast cells.

We totally agree with your proposals. We investigated the effects of dexamethasone on the process of mast cell maturation in vitro and on phenotypic changes of mature cutaneous mast cells in vivo. Because we did not isolate cutaneous mast cells, we could not draw any clear conclusions about the dexamethasone-induced changes of cellular gene expression. It is very important to clarify the effects of glucocorticoids on mature connective tissue-type mast cells, such as peritoneal mast cells. Because the proposed study should contribute to understanding the clinical impact of glucocorticoids on tissue mast cells, we will try to investigate it in the following research.

 3.  To examine the effects of specific knockdown of Mrgpra4, Mrgprb1, Mrgprb2, Mrgprb10, Mrgpre, Gai1, Gai2, or Gai3 on compound 48/80-mediated degranulation of CTMC-like MC generated by the authors will lead you to make a definite conclusion.

We totally agree with your proposals. However, it is technically difficult for us to perform reproducible gene transfer into bone marrow-derived cells. Retroviral gene transfer is available but we could not obtain reproducible results using it. Using the gene-targeted mice is another way to determine the functions of these genes, although such mouse strains are now not available for us. Because the proposed experiments will contribute to more precise understanding of the roles of these genes in mast cells, we would like to perform them in the following research.

Round  2

Reviewer 1 Report

No comments